# Tourism Related to Aromatic and Medicinal Plants: Some Practical Evidence

João Capucho [1,*], Arminda do Paço [1] and Pedro Dinis Gaspar [2]

1 NECE—Research Centre for Business Sciences, University of Beira Interior, 6200-001 Covilhã, Portugal; apaco@ubi.pt
2 C-MAST—Center for Mechanical and Aerospace Science and Technologies, University of Beira Interior, 6200-001 Covilhã, Portugal; dinis@ubi.pt
* Correspondence: joao.capucho@ubi.pt

**Abstract:** This study examines the dynamic interaction between tourism and the Aromatic and Medicinal Plant (AMP) industry in Portugal. An innovative questionnaire was methodically designed, covering a wide range of topics such as pharmacy, tourism, economics, the environment, and circularity. A dataset of 25 separate farmers' responses was produced by focusing specifically on responses within the tourist sphere. Particularly, six of these participants engaged in tourist activities in addition to regular AMP efforts. The questionnaire, which is organized around three open-ended questions, yields critical information. It identifies the major motivations driving farmers toward more sustainable strategies in their tourism-focused AMP operations. Furthermore, it identifies the major barriers to tourist growth in the AMP business. Finally, it outlines the benefits of more closely connecting the tourism sector and AMP enterprises. This study closes with an in-depth discussion of its most significant implications and conclusions, providing useful insight for industry stakeholders and academics alike.

**Keywords:** tourism; Aromatic and Medicinal Plants (AMPs); farmers; sustainability; economics

## 1. Introduction

The tourism sector is a key contributor to the economies of many countries worldwide. In fact, according to the World Travel & Tourism Council (WTTC), the tourism sector accounted for 10.3% (USD 9.630 billion) of the world economy's GDP in 2019 [1]. In the specific case of Aromatic and Medicinal Plants (AMPs), it is still a flourishing but promising segment of the tourism sector.

AMPs play a pivotal role in bolstering both local and national economies, simultaneously sustaining numerous rural livelihoods. They are integral to several industries, encompassing cosmetics, pharmaceuticals, and food. Moreover, they are a product that could be explored in the tourism sector, especially in nature and experience tourism. Thus, AMPs and AMP-based products can be employed for everything, from decorating hotel rooms and providing natural remedies for guests to helping boost local economies through sustainable tourism initiatives. Therefore, countries or regions that choose to work with and introduce the AMP market in tourism can gain competitive and comparative advantages over other countries or regions [2].

Another aspect is that AMPs are a vital element for the sustainability of forests and contribute significantly to countries' economic growth. In this sense, it is essential to expand this market and foster consumer demand for these natural products [3]. According to the World Bank, world trade in AMPs is expected to reach USD 5 trillion by 2050 [4].

There is a demand trend that leverages the AMP sector because of the number of opportunities to create companies and ventures based on planting this raw material, or developing products based on AMPs (medicinal products, perfumes, and cosmetics, among others) [2]. However, although there are several opportunities for entrepreneurs in this

sector, they are aware of the uncertainties and challenges, namely, the ones related to the lack of information about AMPs, and their market, supply chain, and trading systems, among others [5].

AMPs can boost the most rural/uninhabited and low-income areas to evolve and grow on several levels, including the economic dimension [6]. The resident community plays a crucial role in the growth and development of a tourist destination [7]. And when there is efficiency among stakeholders at the economic, social, and environmental levels, it is possible to provide significant advantages to tourist destinations [8]. The AMP sector is exciting and challenging because it can create jobs, sustain economic development, and increase the purchasing power of producers.

It was also found that the mastery of AMPs was helpful when the coronavirus (COVID-19) pandemic appeared because of the lack of drugs and vaccines against this virus in the initial phase. In this follow-up, AMPs were considered a possibility to assist, prevent, and mitigate the severe respiratory infections caused by the COVID-19 virus [9].

The study developed by Okosodo and Mohapatra analyzed and investigated the application of AMPs for treating malaria and typhoid fever and the advances in the field of AMPs as a health tourism product. This research involved interviews with several stakeholders, and it was concluded that a great proportion of them considered APMs to promote the tourism sector; it was also found that 81 different species of AMP were simultaneously employed as a health tourism product and in the treatment of malaria and typhoid [10].

Tourism, over the past half-century, has significantly augmented global wealth [11]. Integrating AMPs can further accentuate this growth, ensuring quality offerings and catalyzing business opportunities, especially in marginalized regions [10]. However, this field of research needs studies and investigations that provide essential insights for the tourism sector to develop and grow coherently and sustainably. Ref. [2] stresses that the AMP sector can bring several benefits, such as the creation of new business opportunities in the tourism sector, and thus create added value in deprived regions.

The knowledge and use of plant species in the local community is still an important part of its life and culture. In this regard, in the study developed by [12], it was concluded that harvesting AMPs in Piatra Craiului National Park in Romania is more vital for the nutrition of local communities than the possibility of creating new jobs and/or higher incomes.

This paper explored the potential of AMPs in the tourism sector, focusing on how they can still be harnessed to create more sustainable tourism initiatives. Questionnaires were used to gather information on the sector and the AMP tourism-related businesses in Portugal. The results were analyzed to identify a few applicable conclusions for politicians, researchers, and entrepreneurs.

## 2. Description of AMPs and their Benefits

Plants known as AMPs have been employed for thousands of years for their medicinal and aromatic characteristics. They are becoming increasingly popular as a natural approach to curing a variety of ailments and enhancing general health. There are several forms of AMPs, each one offering its own set of advantages. Lavender, chamomile, and peppermint are among the most popular AMPs. These plants may be used to make essential oils, drinks, and therapies, among other things. Furthermore, AMPs are raw materials from nature with high added value and are thought to be critical for a variety of sectors, including the protection of forest resources and biodiversity [2].

AMPs have been present since the earliest days of humankind [2,5,13–15]. And over the last few years, this sector has started to play an increasingly important role in creating sources of income and jobs for different ethnic groups and distinct cultures [16]. These resources are full of history and culture and are raw materials that can still create new business projects and ventures based on biodiversity and natural resources. It should

also be noted that these plants are deeply rooted in Mediterranean cultural heritage and spiritual life [17,18].

Typically, the AMP sector boasts several complexities, i.e., it requires the harvesting, processing, packaging, and selling of these plants to the end consumers [3]. The AMP sector encompasses a spectrum of stages, from harvest to market. These stages include harvesting, processing, trading, value addition, and product innovation [2].

The workers in the AMP sector are fundamentally recruited from the local population, and small businesses have the know-how of various production, harvesting, drying, and distillation techniques for these types of raw materials. Ref. [3] clarifies that an AMP's geographical origin and cultural value are determining factors in consumers' purchasing decisions. For example, the Himalayan region in India presents a diversity of AMPs due to its topography and climatic conditions. However, the excessive use of natural resources, climate change, and increasing tourism activities can negatively impact this sector [16]. However, alpine ecosystems are essential for economic development and the population's quality of life, providing the population with AMPs, varied public services, drinking water, food, and energy, among others [19].

Innovation capacity is important in forestry, particularly in AMPs, and can help reduce poverty levels and other associated problems [2]. However, production costs are a constraint for small producers [6]. It should also be noted that fluctuations in plant prices and unfamiliarity with the trading system discourage producers and employees in this sector. Thus, the uncertainties and volatility of the AMP sector have led to a decrease in expectations regarding the stability of markets, causing instability and insecurity for producers [2].

There is strong evidence of the value and potential of AMPs around the world [3]. AMPs offer a wide range of benefits that the tourism sector can leverage.

The AMP industry has high added value, and its products are used in various industries, including cosmetics, pharmaceuticals, and food. Aromatherapy, which uses essential oils extracted from AMPs, is one of the most popular complementary therapies in the world. It can help with various conditions, including stress, anxiety, insomnia, and pain relief. In this regard, several people are now looking for holidays that offer opportunities to learn and experience the mastery of AMPs. For example, there is a growing number of plant-based retreats where guests can learn about the benefits of these amazing plants. In this way, incorporating AMPs into the tourism sector can help boost the economy while promoting the well-being of tourists and the community.

Flouchi and Fikri-Benbrahim defend that some AMPs are effectively used to prevent the risk of contamination and treat some symptoms of COVID-19, such as (i) quinquina; (ii) eucalyptus; (iii) thyme; and (iv) artemisia, among others. These plants assume a very important role since they include bioactive constituents that can be applied in developing new drugs for the epidemic caused by the SARS-CoV-2 virus. It should also be noted that these drugs based on AMPs are examples of drugs with minimal or no adverse effects [9].

AMPs can bring economic benefits to regions of cultivation. In some cases, they are rural and poor areas, and the AMP sector is called to assist by helping to reduce poverty levels and socio-economic problems [2]. In various studies [2,14,20], rural and poor communities have applied AMPs in their healthcare, which is their main income source.

In regions where forest areas exist, AMPs contribute significantly to wealth creation and opportunities for community households [3]. Thus, the AMP sector is vital for the livelihoods of local people and the creation of new jobs, especially for women and young workers [3]. This sector has led to several benefits for various Mediterranean communities in different businesses, such as (i) medicines; (ii) food flavorings; and (iii) beauty and cosmetics. These plants are linked to the evolution of traditional knowledge and cultural heritage [21,22].

The development of mountainous areas is based on the application of local resources sustainably. However, with the population shifting from rural to urban areas, the customs,

traditions, techniques, and know-how related to the AMP sector lose some relevance in terms of cultural and environmental heritage [23].

The study developed in the Piedmont region (Italy) by [6] concluded that most of their production is located in the plains (89% of production), and only 6% and 5% of production is carried out in mountain and hill areas, respectively. However, most of the raw material in the plains is processed to produce drinks and liqueurs. On the other hand, in the more mountainous areas, the species produced are mostly sold as a final product in local shops, herbalists, cooperatives, and restaurants. The same authors concluded that producers in hill and mountain areas boast high interest in this sector of AMPs, as it is a sector that can grow and expand due to its opportunities. However, they report a lack of information and new practical techniques that could be provided to producers in this sector [6].

Concerning the sustainable harvesting of AMPs from nature, using technical and practical knowledge, and their subsequently marketing as unprocessed raw materials, it is framed as being a way of promoting sustainable economic development [10]. Ref. [2] argues that if marketing channels are efficient, cultivating and harvesting AMPs can be economically viable, i.e., be a source of income for various entrepreneurs and business owners.

## 3. The Tourism Sector and AMPs

The tourism sector is a key driver of the economy in many countries worldwide. AMPs are a valuable resource for the tourism sector, providing a unique selling point and attracting visitors from around the world. It also has the potential to create jobs and generate income for local communities. There are many examples of how AMPs are being used in the tourism sector, from hotels and spas using them in their products and treatments to tour operators offering guided tours of AMP plantations.

As part of the Mediterranean heritage, AMPs are a product crucial for tourism in several areas, representing a product with high added value [3]. The use of AMPs in tourism is an exciting new trend with great potential. With the right policies and investments, AMPs can significantly contribute to developing sustainable tourism initiatives.

Countries based in the Mediterranean benefit from the agroclimatic conditions for the production, treatment, and harvesting of AMPs, and the traditions and culture that are passed on across generations [3]. Thus, customs, techniques, and traditions that sustain the identity of rural communities, and can provide a point of interest to tourists, should be protected [23].

A project developed in Trentino (Italy) applied the sustainable use of naturally grown AMPs in rural areas. The AMP sector was selected for this project and this rural area due to the strong positive impacts that can originate from the revitalization of economies, business investments, and the sharing and creation of culture and knowledge. The project also developed a tourism pack to foster a tourism culture based on AMPs, i.e., cooperation between AMPs growers, sustainable harvesting, and the tourism sector [23].

Regarding the creation of strong national brands, they can positively affect the country at the level of international trade [24]. Therefore, positive impacts can arise from different socio-economic determinants where tourists enjoy the experience of travelling [7,25]. In this sense, for the sake of developing a positive and robust identity of a tourist destination, businesses, residents, local managers, and competent authorities, mainly at the institutional level, should create joint marketing strategies to promote the creation of consistent and innovative brands [7,26].

### 3.1. AMPs and the Hospitality Industry

AMPs have been used in the hospitality industry for many years. AMPs can be used to make natural soaps, perfumes, and cosmetics. They can also be used to create aromatherapy products that can be used in spas and hotels. In addition, many AMPs have medicinal properties that the hospitality industry can use to treat guests with ailments such as stress, anxiety, and insomnia.

There are a few reasons why the hospitality industry should continue to use AMPs. They are, first and foremost, a natural and ecological resource. They will not contaminate the environment or drain resources, unlike synthetic materials. Furthermore, they are frequently more successful than synthetic materials in curing the ailments of guests. Finally, the usage of AMPs can aid in the production of greater rates of business growth.

Due to the low cost of these natural remedies and the lack of adverse effects, AMP-based medications attract a large number of visitors and patients. "Herbal tourism" refers to tourism that involves the use of AMPs in medications or therapies [2].

Employing AMPs in tourism is an excellent method to provide additional peacefulness and pleasure for visitors. There are several ways to include AMPs in a company, ranging from utilizing them in spa therapies and aromatherapy to using them as a component of an agricultural or cooking experience.

The tourist industry is starting to see AMPs' potential to attract people and help the economy. As part of their spa services, several hotels and resorts now provide AMP-based therapies. Furthermore, an increasing number of travel companies are offering AMP-themed excursions, letting guests discover more about these plants and the best ways they can employ them to enhance their health and well-being.

One of the most common applications of AMPs in tourism is in spa and wellness treatments. Aromatherapy, for example, is a massage that employs essential oils from plants to encourage relaxation and well-being. Another way to use AMPs in the tourism sector is to offer them as part of a farming/gardening or cooking experience. Many culinary herbs, such as basil, oregano, and thyme, are also great medicinal plants. Guests can learn about the history and uses of these herbs while enjoying a delicious meal or creating a cup of herbal tea, which is a way to add interest and relaxation for guests.

### 3.2. Opportunities and Critical Factors for the Implementation of the AMP-Related Tourism

According to the World Health Organization, a significant part of the world's population uses AMPs (three billion). The AMP industry is a promising sector with great tourism potential. However, several critical factors need to be addressed to ensure the sustainability of this industry [27].

The growing demand for eco-friendly plant-based products reinvigorates rural economies and improves livelihoods [2]. Consequently, exploitation of the AMP sector has increased significantly and has reached levels considered above sustainable levels, calling into question the sustainability of this sector in the long term [14]. It should be added that in the study developed by Kala, it was shown that the demand for AMPs may lead to higher rates of harvesting of some species, which may result in the loss of some species and genetic biodiversity [14].

One of the key factors is the availability of raw materials. AMPs are generally grown in wild areas rather than under controlled conditions, making it difficult to guarantee a constant supply of raw materials. In addition, harvesting AMPs often requires specialized knowledge and skills, which may not be readily available in all regions and local communities.

The businesses of the AMP market are deeply complex, and it is necessary to develop market analysis and be aware of market trends [3].

The marketing and promotion of AMP products are often challenging due to a lack of consumer knowledge and awareness of these products. There is also a perception that these products are expensive and meant only for luxury markets. A large part of the population in Mediterranean countries consumes products and services based on AMPs. In that sense, they are aware of the need to certify and label products based on these plants to promote more transparency and encourage sustainable crops [3].

AMPs are essential for local development and biodiversity conservation; however, several challenges and knowledge gaps can give rise to failures and jeopardize this sector [3]. Mountain areas are faced with several challenges over time, such as (i) depopulation; (ii) a lack of competitiveness; and (iii) climate change impacts [23].

Innovations in agriculture in the AMP sector can catalyze a new business's success and produce comparative and competitive advantages in this sector [2]. However, the innovative capacity in the AMP sector is limited due to the scarcity of research and development, the limited diversity of the business fabric, and the weak institutional investment of Mediterranean countries in the sector [3].

According to Kala, AMPs in the Alpine region are facing strong anthropogenic threats (exploitation; road construction; and unregulated tourism development) and natural threats (changing weather conditions and the invasion of exotic species) [13].

Climate change, difficulties in harvesting AMPs, and low returns on investments discourage entrepreneurs in the AMP sector. Developments in the AMP sector cause fluctuations and changes in the production orientation of producers, thus combating market needs and opportunities [2].

Tourist destinations can create gastronomic activities (show cooking; restaurant itineraries), fairs, and festivals. Some countries focus on promoting tourism and products related to AMPs. In this sense, an important factor in promoting the AMP sector is the development of national and international fairs and events with the help of territorial marketing strategies. This type of marketing can produce several positive effects in attracting new tourists and encouraging the consumption and use of AMPs and products related to these raw materials [7].

In sum, the growing trend of health and wellness tourism [10] is leading to the creation and development of businesses in the AMP sector.

## 4. Methods

Data were collected as part of the PAM4WELLNESS project in Portugal. This study aimed to contribute to understanding the economic value of the endemic collected species already contributing to the cosmetic and pharmaceutical industry there. At present, the harvesting of these species is primarily aimed at forest clearing and fire prevention, not at extracting the potential of the collected material. In fact, at present, the only value extraction from the harvest is through its dispatch to the biomass industry.

This project also aims to explore the sustainability of the harvesting and use processes. Through a suitable benchmarking exercise, this study aims to identify the best practice for production processes, waste use and treatment, and leveraging other economic benefits such as tourism. All of this is aimed at maximizing economic return while minimizing environmental impacts and reducing the ecological footprint of these processes.

Several producers/farmers were enrolled in this project to collect evidence regarding factors such as tourism, production, pharmaceutics, and circularity, among others. In this specific case, we will focus on tourism issues. The research team formulated a questionnaire, emphasizing open-ended queries to procure in-depth insights. Google Forms was the chosen medium for data collection, complemented by an interview guide to maintain uniformity. The survey attracted participation from 25 farmers from Portugal (Table 1). The small number of farmers surveyed was due to the limitations of the AMP sector in Portugal.

**Table 1.** Number of survey respondents and types of farmers.

| AMPs Farmers | Number of Survey Respondents |
|---|---|
| AMP farmers with no tourism activity associated with their production business | 19 |
| AMP farmers with tourism activities associated with their production business | 6 |
| Total AMP farmers surveyed | 25 |

**Source:** own elaboration.

The farmers were given a questionnaire with several questions related to the specificities of the AMPs produced, and at the end, three open-ended questions related to tourism and the AMP industry (reasons to act sustainably, obstacles, and benefits) were presented.

## 5. Results

The questions and respective answers related specifically to the tourism agents that were simultaneously producers of AMPs were grouped by category as follows:

**Question 1 (Q1).** *What reasons might lead a producer to act more sustainably in the tourism sector linked to AMPs?*

**Question 2 (Q2).** *What are the main barriers/obstacles that could hinder the development of the tourism sector linked to AMPs?*

**Question 3 (Q3).** *What are the main benefits that could foster the development of the tourism sector linked to AMPs?*

The answers obtained, shown in Tables 2–4, were summarized in key points, as shown after in Table 5 below:

**Table 2.** Reasons to be more sustainable.

| Category | | Responses |
|---|---|---|
| Service improvement | - | Selecting AMPs for their spaces as ornamentals, and using them in the preparation of drinks, meals, and even hygiene and cleaning products; |
| | - | Improving the quality of the service provided. |
| Sustainability awareness | - | Encouraging styles of tourism that are beneficial to all species; |
| | - | The basic principle of farmers of AMPs is sustainability; |
| | - | The appreciation of sustainable natural and agricultural heritage by tourists; |
| | - | Sustainable consumption. |
| Marketing/business issues | - | Improving brand image; |
| | - | It can help with dissemination and subsequent marketing; |
| | - | Promote products and increase sales; |
| | - | Economic profitability and customer awareness. |
| Concern with production and service | - | To raise awareness of how our products are produced; |
| | - | Valuing products; |
| | - | Passing on good practices to clients and consumers, to be an informative agent. |
| Others | - | To have more credibility; |
| | - | To have more support. |

**Source:** data collected as part of this study.

**Table 3.** Barriers/obstacles.

| Category | | Responses |
|---|---|---|
| Human resource constraints | - | Lack of workforce trained in tourism; |
| | - | Lack of manpower; |
| | - | Lack of knowledge in the field of tourism. |
| Financial problems | - | Financial support in the field of tourism; |
| | - | High initial investment; |
| | - | Financial power to move forward with projects. |
| Marketing barriers | - | Price and distribution; |
| | - | Lack of size of the market; |
| | - | Poor communication in the sector. |

**Table 3.** *Cont.*

| Category | Responses |
|---|---|
| Regulatory issues | - Too much bureaucracy;<br>- Difficulty in obtaining a building license for tourism. |
| Location problems | - The location of the farm;<br>- Access and facilities;<br>- Interior of the country. |
| Others | - Secrecy;<br>- Cultural issues. |

**Source:** data collected as part of this study.

**Table 4.** Benefits.

| Category | Responses |
|---|---|
| Marketing and market | - Larger target audience;<br>- To raise awareness of our brand and what we do;<br>- Promoting the product and increasing sales;<br>- Raising awareness on site, increasing sales, and building loyalty;<br>- Image and additional profits;<br>- Increases and improves the customer experience;<br>- Dissemination;<br>- Creating differentiation from the competition. |
| Awareness of sustainability | - Making more use of local resources;<br>- Increase sustainable employment in sparsely populated areas;<br>- Improve the balance of the ecosystem. |
| Stimulating the awareness of others | - Raise awareness and stimulate the appreciation of AMPs throughout society;<br>- Raising tourist awareness of the value and protection of nature;<br>- Education on the use of AMPs in food and medicine;<br>- Health and disease prevention. |
| Financial | - Financial benefits;<br>- Economic valorization;<br>- Increases business sustainability;<br>- Source of income. |

**Source:** data collected as part of this study.

**Table 5. Summary of** key points from stakeholders' answers.

| | |
|---|---|
| Reasons to be more sustainable in the tourism sector linked to the AMP industry **(Q1)** | - Service improvement;<br>- Sustainability awareness;<br>- Marketing/business issues;<br>- Concern with production and service. |
| Main barriers/obstacles to the inclusion of the AMP industry in the tourism sector **(Q2)** | - Human resources constraints;<br>- Financial problems;<br>- Marketing barriers;<br>- Regulatory issues;<br>- Location problems. |
| Main benefits obtained from including the AMP industry in the tourism sector **(Q3)** | - Marketing and market;<br>- Awareness of sustainability;<br>- Stimulating the awareness of others;<br>- Financial. |

**Source:** data collected as part of this study.

Concerning the first and third questions posed to farmers, the answers were selected and clustered into four dimensions: (i) environmental; (ii) economic and legal; (iii) social and brand image; and (iv) resilience and health. Figures 1 and 2 show the frequencies of the dimensions in the farmers' answers to questions 1 and 3, respectively.

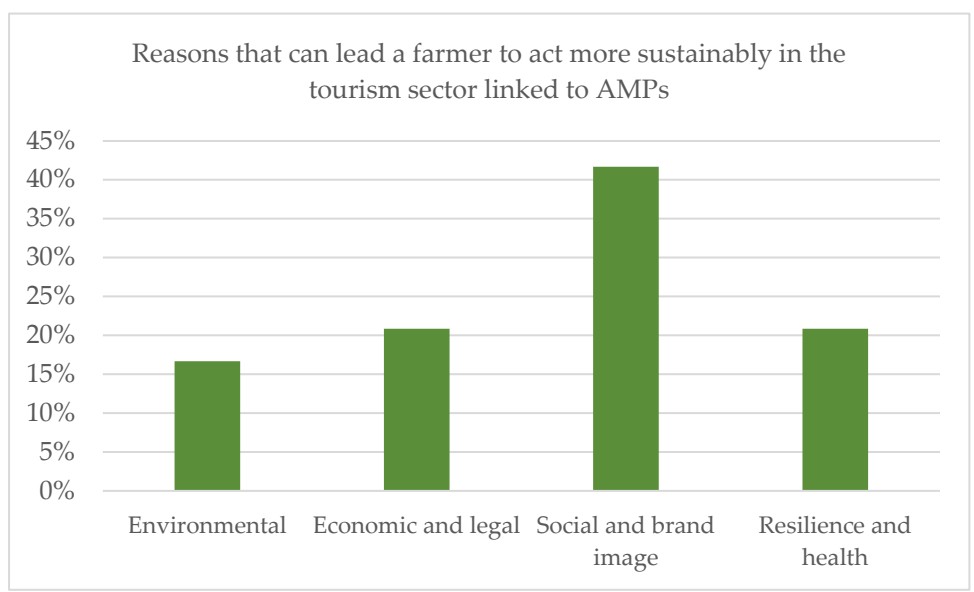

**Figure 1.** Reasons to be more sustainable. **Source:** data collected as part of this study.

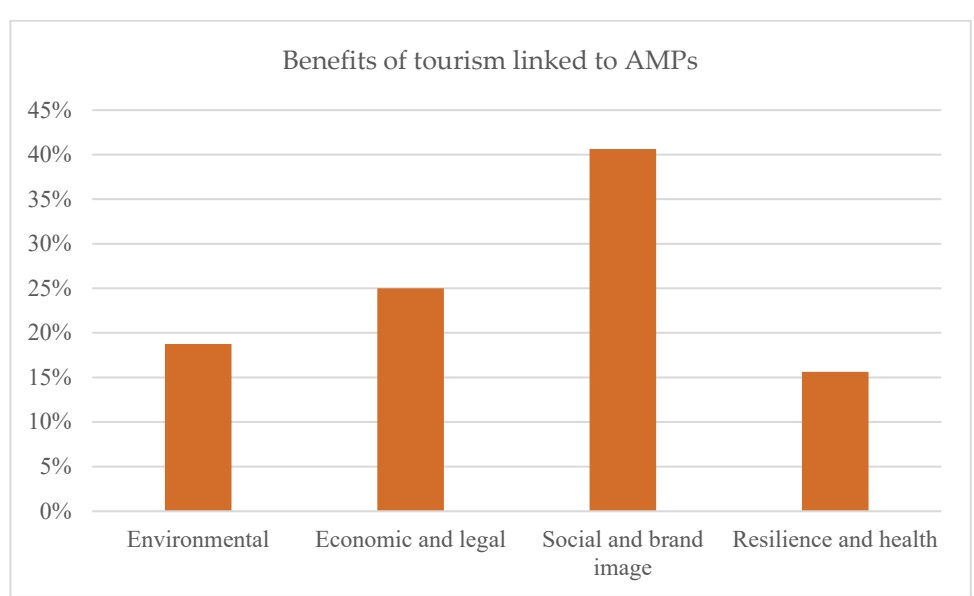

**Figure 2.** Benefits of tourism linked to AMPs. **Source:** data collected as part of this study.

These results indicate that farmers prioritize the social dimension and brand image in both the first and third questions. The results exhibited that farmers see society and brand image as a strong benefit and reason to consider AMPs in their tourism businesses, as well as acting more sustainably in their businesses. Other dimensions are also significant from farmers' perspectives. However, in the first question, both the economic and legal dimensions, as well as resilience and health, hold equal importance at 21%. The environmental dimension stands at 17%. In the last question, the order of frequency of the dimensions was changed. Thus, economic and legal benefits were the two most emphasized dimensions in the answers (25%), followed by environmental benefits (19%), and finally, resilience and health benefits (16%).

Regarding the second question, a graph was not drawn up due to the homogeneity of the answers. The answers were summarized into five key barriers: (i) a lack of knowledge and advancement in the AMP sector; (ii) bureaucracy and a lack of financial support for stakeholders; (iii) a lack of qualified labor; (iv) cultural and communication barriers in the AMP sector; (v) access and geographical location.

## 6. Discussion and Implications

The AMP industry in the tourism sector is an important driver of economic growth and development. However, the ecosystem in which the AMP industry operates does not always foster its growth and development. The tourism-related AMP industry is subject to several policy regimes, including trade, agriculture, and tourism. These policy regimes can often be contradictory, making it difficult for the AMP industry to operate coherently and in a coordinated manner. For example, while tourism may be encouraged by government policies promoting exports, these policies may inadvertently discourage the cultivation of AMPs, making it more difficult for farmers to access export markets. The lack of a clear and coordinated policy framework for the AMP industry can negatively affect investors and workers in the sector. Investors may be deterred from investing in the AMP industry due to uncertainty about future policy changes. At the same time, workers may be forced to accept lower wages or bad working conditions due to a lack of job security.

The AMP sector is developing; however, some gaps and complexities have been identified. In this regard, policymakers and the private sector could cooperate to address these challenges. Creating brands for products and the organizations (e.g., cooperatives, associations) is one of the main challenges of this market. Therefore, it is necessary to work hard on the branding and certification of products to achieve the valorization and development of this sector [3].

According to Ref. [3], several measures related to the certification of AMPs may be introduced, such as (i) forest management certification; (ii) social certification; (iii) organic certification; (iv) product quality certification; and (v) certification of origin, among others. These measures that aim to promote certification are relevant to encouraging a sustainable culture of forest resources, to promoting better working conditions and well-being for workers, and to obtaining improved products. Kala stressed the importance of developing the certification of AMP to help entrepreneurs and business owners in this sector and improve consumer confidence indices [2]. Rovira, Garay, Górriz-Mifsud, and Bonet mentioned that certification and cooperation between stakeholders are crucial for developing tourism destinations and products [7].

The marketing and commercial strategy can be adapted to the consumers' purchasing trends, balancing the high average production costs and the price fluctuations driven by the uncertainties of this market. In this sense, innovative marketing approaches should be fostered to match the buyers' perspectives (design, packaging, and distribution channels, among others) [3].

Countries must create effective policies that promote the harmonization of harvesting by supporting the protection of wild plants. Misaligned and inappropriate practices in the exploitation of AMPs can give rise to negative impacts on the development of this sector. In this sense, the management of AMPs presents several challenges due to the depletion of natural resources, overexploitation, unsustainable harvesting practices, and the misuse of forest lands [2]. It should also be noted that wild AMP harvesting organizations (e.g., associations, cooperatives, and research and development groups) could disseminate and shares among all stakeholders [3].

Governments could implement a measure that sets a guaranteed minimum price to promote confidence and contribute to the emergence of new producers in the AMP sector [2]. It is also important to develop local economies to adopt policies based on sustainability, innovation, and cultural enhancement [23].

Overall, the AMP sector lacks research on innovative ideas to boost and energize this industry. Thus, one measure that could be established for developing the AMP

sector is creating a communication network between stakeholders to share information and techniques on the cultivation and harvesting of these AMPs in a sustainable and up-to-date manner. Policymakers and stakeholders could cooperate to promote technical courses/training/workshops to foster knowledge sharing and technical know-how on AMPs. These training types can positively affect entrepreneurs' knowledge at the practical level and at the business management level, and also in the creation of business measures and strategies [2].

Additionally, the public and private sectors could cooperate to enhance the cultural heritage and traditions of the regions, with positive consequences at several levels: (i) cooperation between the local community and regional producers (micro-production, incentives for investment and production); (ii) entertaining activities on AMP sites; (iii) the expansion of tourism offerings (several itineraries and packages of innovative tourism experiences); (iv) the creation of small plots where tourists could experience planting and the culture and traditions of rural localities; and (v) cooperation between schools, institutions, and the industrial sector [23].

Another measure that could be implemented is the creation of community gardens in rural areas based on the cultivation of AMPs to enhance cultural understanding, education, and learning about this type of cultivation. It would be beneficial to establish a comprehensive management plan. Additionally, creating training and education centers that offer techniques for producers and entrepreneurs in the AMP sector can help mitigate negative impacts, such as overexploitation, deforestation, and other environmental challenges [10].

The Italian government funded a project in Piedmont with the intention of determining the status of AMPs in this region in terms of the cultivated area, the particularities of the farmer/producer, and the organization of the production chain, through information gathering, data analysis, and the creation of data on the impact AMPs on the sector in the region [6]. In this sense, it is suggested that the Portuguese government invest in incentive projects and measures to boost this sector, along with establishing communication and territorial marketing strategies to promote these regions.

## 7. Conclusions

This study aims to highlight the current state of advancement in the AMP sector and its relationship with the tourism sector and further identify limitations and future challenges that may be vital for this sector's sustainability, innovation, and growth.

AMPs have great potential for the tourism sector, i.e., they can be used to attract tourists, create products and services, and generate employment opportunities. In addition, the AMP sector can also help improve local communities' quality of life and be a great opportunity for sustainable development in the tourism sector.

Overexploitation and unsustainable harvesting seem to be some of the main challenges in this sector [3]. Thus, the main threats and problems related to AMP resources should be analyzed with the stakeholders, and the constraints in production, financing, and marketing in the AMP sector should be investigated and identified.

It should be noted that there is limited information regarding habitat suitability, climate change impacts, management strategies, and monitoring of the AMP sector. In this regard, it is necessary to analyze these aspects in future research [16].

According to the study developed in [3], the challenges for the AMP sector are as follows: (i) the development of the AMP market; (ii) quality of life and well-being; (iii) research and development; (iv) product certification and labeling; and (v) new ideas and techniques for AMPs processing and treatment.

The success of enterprises in the AMP sector is strongly correlated with training and information on new techniques. Therefore, the more information producers and entrepreneurs can capture, the higher their success rate will be. Therefore, training and scientific/technical events can positively impact yields and the productive efficiency of plantations [2].

The main results obtained regarding the barriers faced by the farmers surveyed were as follows: (i) a lack of knowledge and advancement in the AMP sector; (ii) bureaucracy and a lack of financial support for stakeholders; (iii) a lack of qualified labor; (iv) cultural and communication barriers in the AMP sector; (v) access and geographical location.

This research, which was carried out in Portugal between April and June 2023, provides an important insight into the views and behaviors of AMPs producers. However, it is essential to recognize its limitations. Due to the narrow geographical focus on Portugal, the results may not be immediately transferable to other locations with different socio-economic and environmental contexts. The study's conclusions may be limited due to the small sample size, which reflects the young nature of the AMP business in Portugal. Furthermore, the sample's specialized character, consisting of farmers, precludes the viewpoints of other important players in the tourism business, such as tourists or professionals in the larger tourism sector. These factors are critical in evaluating the study's findings and should be considered when extending them to larger settings or making policy suggestions.

The lack of data/databases is a constraint for policymakers who wish to undertake analysis and interventions in biodiversity conservation. Thus, this limitation should be addressed. Therefore, new research on this subject should be pursued, which would also assist in the development of an appropriate research agenda [16].

Ref. [16] add that there is also a lack of research regarding AMPs' role in producing energy through biomass. Thus, two future research proposals are suggested: (i) research to promote industrial symbiosis between AMP industries and the tourism sector, and (ii) studies that apply circular economy domains in AMP industries and the tourism sector (e.g., using surplus AMPs as a source of biomass energy production in small-scale power stations in hotels).

**Author Contributions:** Conceptualization, J.C. and A.d.P.; methods, J.C. and A.d.P.; formal analysis, J.C. and A.d.P.; investigation J.C.; resources, J.C., A.d.P., and P.D.G.; data curation, J.C. and A.d.P.; writing—original draft preparation, J.C. and A.d.P.; writing—review and editing, A.d.P. and P.D.G.; supervision, A.d.P. and P.D.G.; project administration, A.d.P. All authors have read and agreed to the published version of the manuscript.

**Funding:** This research was funded in part by the Fundação para a Ciência e Tecnologia (FCT), C-MAST, under project UIDB/00151/2020; NECE-UBI under project UIDB/04630/2020; a Ph.D. fellowship (2023.00312.BD); and project POCI-01-0246-FEDER-181319 (PAM4Wellness), financed by the European Regional Development Fund (ERDF) of the European Union through POCI—the Competitiveness and Internationalization Operational Program.

**Institutional Review Board Statement:** Not applicable.

**Informed Consent Statement:** Informed consent was obtained from all subjects involved in this study.

**Data Availability Statement:** The data presented in this study are available on request from the corresponding author. The data are not publicly available due to privacy or ethical restrictions.

**Acknowledgments:** We acknowledge all the participants of this study.

**Conflicts of Interest:** The authors declare no conflict of interest.

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
