# Peer review of "Tourism Related to Aromatic and Medicinal Plants: Some Practical Evidence"

_sustainability, doi:10.3390/su152215966_

Round 1
Reviewer 1 Report
Comments and Suggestions for Authors
- The research background discussion on AMP and tourism is quite connected and important. However, if we focus on the research purpose, we cannot find the opinions of tourism-related industries, and only the opinions of 25 farmers are identified.
- In addition, where is the PAM4WELLNESS data used? Is it helpful for relevant explanations in this article?
- The research results show qualitative questions and answers. This part uses a Google online questionnaire. Please explain the reliability and validity of the questionnaire processing.
- Where did the questionnaire questions come from? How was the structure of these questionnaires generated? Is there relevant literature to support your topic?
- The study divided farmers into groups engaged in tourism and farmers not involved in tourism, but the results of the grouping were not explained in the research results.
- How are the four groups in Figure 1 and Figure 2 defined and divided into groups? I don’t know how to connect them to the above. In addition, the data in the figure is the percentage of farmers’ responses. How to summarize these four groups to produce the percentages in Figure 1 and Figure 2?
- The discussion in the conclusion needs to provide a substantial link to the results.
English writing requires more academic text to express the explanation of the article.
Author Response
Comments and suggestions from the first reviewer:
- “The research background discussion on AMP and tourism is quite connected and important. However, if we focus on the research purpose, we cannot find the opinions of tourism-related industries, and only the opinions of 25 farmers are identified.”
We sincerely thank you for the time and effort you put into reviewing our article.
This study was based on the development of a questionnaire only for farmers of Aromatic and Medicinal Plants, a growing and still small sector in Portugal. We only received 25 responses, six of which are linked to tourism activities in their business. The industrial part was developed within this project by a different team.
We suggest that future research could further explore the opinions of companies and stakeholders in the tourism sector for a more comprehensive understanding.
- “In addition, where is the PAM4WELLNESS data used? Is it helpful for relevant explanations in this article?”
We understand the concern raised by the reviewer on PAM4Wellness project data. We would like to clarify that this study is based on a content analysis, where the results are developed according to the open answers developed in the project's final questionnaire. This questionnaire covered various areas, such as tourism, engineering, pharmaceuticals, among others.
- “The research results show qualitative questions and answers. This part uses a Google online questionnaire. Please explain the reliability and validity of the questionnaire processing.”
We understand the concerns expressed about the reliability and validity of the questionnaire. We stress that the questionnaire and the results were analysed in detail and checked for consistency with the existing literature.
- “Where did the questionnaire questions come from? How was the structure of these questionnaires generated? Is there relevant literature to support your topic?”
We understand the importance of clarifying the development of the questions in the questionnaire.
The questions in the questionnaire were developed based on a review of the literature related to the topic in question. The structure of the questionnaire was carefully designed to capture farmers' perceptions in a comprehensive manner.
The structure of these questionnaires was developed by several teams, and our team developed its part with open-ended questions with the aim of analysing qualitatively and with the approach of a content analysis to offer an innovative and yet unstudied view in this field of research. The relevant literature to support this research topic was mainly these four articles:
- Taghouti, I.; Cristobal, R.; Brenko, A.; Stara, K.; Markos, N.; Chapelet, B.; Hamrouni, L.; Buršić, D.; Bonet, J.A. The Market Evolution of Medicinal and Aromatic Plants: A Global Supply Chain Analysis and an Application of the Delphi Method in the Mediterranean Area. Forests 2022, 13, 1–13, doi:10.3390/f13050808;
- Kala, C.P. Medicinal and Aromatic Plants: Boon for Enterprise Development. J Appl Res Med Aromat Plants 2015, 2, 134–139, doi:10.1016/j.jarmap.2015.05.002;
- Kala, C.P. Commercial Exploitation and Conservation Status of High Value Medicinal Plants across the Borderline of India and Nepal in Pithoragarh. Indian Forester 2003, 129, 80–84, doi:10.36808/if/2003/v129i1/2238;
- Nicola, S.; Hoeberecht, J.; Fontana, E.; Saglietti, D. Medicinal and Aromatic Plants in Italy: Situation and Perspective for the Piedmont Region. Acta Hortic 2004, 629, 375–382, doi:10.17660/ActaHortic.2004.629.48.
- “The study divided farmers into groups engaged in tourism and farmers not involved in tourism, but the results of the grouping were not explained in the research results.”
We understand the concern raised by the reviewer about the farmers' results. We would like to clarify that the answers were only provided by farmers who have activities linked to tourism. In other words, the content analysis of the answers obtained are from the six producers who have their aromatic and medicinal plant business linked to the tourism sector. The remaining producers did not answer these questions.
- “How are the four groups in Figure 1 and Figure 2 defined and divided into groups? I don’t know how to connect them to the above. In addition, the data in the figure is the percentage of farmers’ responses. How to summarize these four groups to produce the percentages in Figure 1 and Figure 2?”
We understand the need for a more detailed explanation of the definition and formation of the four groups shown in the figures. Regarding to the point raised in Figures 1 and 2, it should be noted that the four groups in Figure 1 and Figure 2 are defined and divided into groups according to the farmers' responses. The percentage is the frequency of the dimension in the farmers' response.
- “The discussion in the conclusion needs to provide a substantial link to the results.”
We recognise the importance of a solid link between the discussion and the results. In the sixth and seventh paragraphs of the conclusion we can see the link between the results obtained and the conclusions of the study: “The success of enterprises in the AMPs sector is strongly correlated with training and information on new techniques. Therefore, the more information producers and entrepreneurs can capture, the higher their success rate will be. Therefore, training and scientific/technical events can positively impact yields and the productive efficiency of plantations [1].
The main results obtained regarding the barriers confronted by the farmers surveyed were: (i) Lack of knowledge and advances in the AMP sector; (ii) Bureaucracy and lack of financial support for stakeholders; (iii) Lack of qualified labour; (iv) Cultural and communication barriers in the AMP sector; (v) Access and geographical location.”
Since the lack of studies related with this specific field it was difficult to discuss all the obtained results.
Reviewer 2 Report
Comments and Suggestions for Authors
Very interesting topic, exploring this relationship between AMP and tourism, contributing with the state of the art and an empirical approach that justifies the relationship and also raises major questions about the best approach to pursuing it.
The big issue that I think it can be better contextualized is the territorial approach that is not in-depth in terms of characterization of respondents, territory of operation and its specificities.
It is mentioned several times that this is a portuguese approach but it is not understood whether the approach is local, regional or national with an effect on the results. What is the representativeness and specificities of the study without more data from the agents who were interviewed?
I would accept the article because of its interest but I think that the results could be strengthened with this territorial and sample specification.
Author Response
Comments and suggestions from the second reviewer:
- “Very interesting topic, exploring this relationship between AMP and tourism, contributing with the state of the art and an empirical approach that justifies the relationship and also raises major questions about the best approach to pursuing it. The big issue that I think it can be better contextualized is the territorial approach that is not in-depth in terms of characterization of respondents, territory of operation and its specificities.”
We would like to express our appreciation for the review and the valuable comments and suggestions provided.
Concerning to the point raised about the territorial approach, we would like to clarify that we tried to cover mainly the territory where the cultivation of AMP is more developed (North, Centre and Alentejo regions). Given the fact of the questionnaire be anonymous we opted to not mention their location. Some would be immediately recognized given theirs impact in the region/location.
- “It is mentioned several times that this is a portuguese approach but it is not understood whether the approach is local, regional or national with an effect on the results. What is the representativeness and specificities of the study without more data from the agents who were interviewed?”
We appreciate your question. The approach is for national producers in various regions of Portugal (NUTS II and NUTS III). This specific sector is recent and has a little number of agents operating in the tourism of AMP.
- “I would accept the article because of its interest but I think that the results could be strengthened with this territorial and sample specification.”
Thank you very much for the time and effort you put into reviewing our article and thank you for your interest in accepting our article for publication.
As mentioned above, the suggested questions to be included in the conclusion cannot be mentioned for anonymous reasons (the sample is small, thus the agents could be identified).
Reviewer 3 Report
Comments and Suggestions for Authors
I appreciate the authors contribution to studying the current state of relations and advances of the AMPs sector with the tourism sector and further identify limitations and future challenges that may be vital for this sector's sustainability, innovation, and growth.
The paper is written with adequate clarity, presents the contributions to the literature and technically is OK. The methodology is well presented, and also, the results are well presented and discussed.
Besides of these, I have some recommendations:
- to present in the Conclusions the policy implications research, practice and society;
- some recommendations to all regulators should be added in order to explain how they can benefit from the findings.
A weak point of the paper may be considered the small number of companies included in the sample which reduces the capacity of generalisation of the study.
Comments on the Quality of English LanguageModerate editing of English language required.
Author Response
Comments and suggestions from the third reviewer:
- “I appreciate the authors contribution to studying the current state of relations and advances of the AMPs sector with the tourism sector and further identify limitations and future challenges that may be vital for this sector's sustainability, innovation, and growth.”
We sincerely thank you for your positive and encouraging feedback. Your words are very motivating and appreciated.
- “The paper is written with adequate clarity, presents the contributions to the literature and technically is OK. The methodology is well presented, and also, the results are well presented and discussed.”
We're grateful for your positive comment. It's gratifying to know that our work has been well received.
- “Besides of these, I have some recommendations:
- to present in the Conclusions the policy implications research, practice and society;
- some recommendations to all regulators should be added in order to explain how they can benefit from the findings.”
We sincerely thank you for your suggestions, however, the penultimate section of the paper "Discussion and implication" refers to various policy implications, practice and society; and some recommendations.
- “A weak point of the paper may be considered the small number of companies included in the sample which reduces the capacity of generalisation of the study.”
We agree with your comment that one of the limitations of this study is the small number of farmers. This is still a small and growing sector. Therefore, we believe that there is still a necessity to study and analyse the link between aromatic and medicinal plant farmers who are connected to the tourism sector.
Reviewer 4 Report
Comments and Suggestions for Authors
The manuscript delves into the potential of Medicinal and Aromatic Plants (AMP) within the realm of sustainable tourism, with a specific focus on mountainous regions. The author commendably underscores the socio-economic and environmental ramifications of AMPs, particularly against the backdrop of rural-to-urban migration and the recent Covid-19 pandemic. Nonetheless, the paper leans more towards a practical report than an academic article due to the absence of a clear development of research questions or hypotheses grounded in prior research.
1. Introduction and Background:
- The manuscript offers a robust foundation on the importance of AMPs in sustainable tourism. Yet, a more exhaustive literature review would amplify the paper's depth.
- Clarity and Flow: Some segments are protracted and could benefit from brevity and restructuring for enhanced clarity.
Suggested Revision: "AMPs play a pivotal role in bolstering both local and national economies, simultaneously sustaining numerous rural livelihoods. They are integral to several industries, encompassing cosmetics, pharmaceuticals, and food."
"The AMP sector encompasses a spectrum of stages, from harvest to market. These stages encapsulate harvesting, processing, trading, value addition, and product innovation."
- Consolidate Information: There's a noticeable repetition and dispersion of certain points. Grouping related information would enhance coherence.
Suggested Revision: "Tourism, over the past half-century, has significantly augmented global wealth. Integrating AMPs can further accentuate this growth, ensuring quality offerings and catalyzing business opportunities, especially in marginalized regions."
2. Methodology:
- The methodology segment appears elusive or inadequately detailed. A comprehensive outline detailing research methods, data collection techniques, and analytical procedures is imperative.
- Detail the Data Collection Process: A meticulous breakdown of the data collection process is essential.
Suggested Revision:
- "The research team formulated a questionnaire, emphasizing open-ended queries to procure in-depth insights."
- "Google Forms was the chosen medium for data collection, complemented by an interview guide to maintain uniformity."
- "The survey witnessed participation from 25 producers/farmers."
- Elaborate on the Questionnaire**: The questionnaire, especially its open-ended components, requires further elaboration. The genesis of the survey questions and their sources should be elucidated.
3. Results and Discussion:
- The manuscript sheds light on AMPs' potential in revitalizing rural and economically challenged areas. The reference to AMPs during the Covid-19 pandemic is particularly captivating. Nonetheless, the results section would greatly benefit from a more organized data representation, preferably through graphs or tables.
- Question 3 (Q3), which reads, ‘What are the main barriers/obstacles that could hinder the development of the tourism sector linked to the AMPs?’ should be reframed to ‘What are the main benefits that could foster the development of the tourism sector linked to the AMPs?’
- Address Stakeholder Perspectives: The discussion should encompass the implications of the findings for diverse stakeholders, including farmers, policymakers, and investors.
The paper, given its topical relevance and insights, certainly has potential. However, it necessitates revisions in line with the aforementioned comments prior to being deemed fit for publication.
Comments on the Quality of English Language
The manuscript is generally well-composed, but a meticulous proofreading is advised to bolster clarity and coherence.
Page 11:
1. "Another measure that could be implemented is the creation of community gardens in rural areas based on the cultivation of AMPs to enhance cultural understanding, education, and learning about this type of cultivation."
- Reviewer's Note: The term "to foster a more efficient culture" was ambiguous. The revised sentence provides clearer intent.
2. "It would be beneficial to establish a comprehensive management plan. Additionally, creating training and education centers that offer techniques for producers and entrepreneurs in the AMP sector can help mitigate negative impacts, such as overexploitation, deforestation, and other environmental challenges [10]."
- Reviewer's Note: The original sentence structure was intricate and potentially perplexing. The revised version breaks it down for enhanced clarity.
Page 10:
3. "The results indicate that farmers prioritize the social dimension and brand image in both the first and third questions."
- Reviewer's Note: The repetitive use of "dimension" made the original sentence somewhat tedious. The revised sentence is more concise.
4. "Other dimensions are also significant from the farmers' perspective. However, in the first question, both the economic and legal dimensions, as well as resilience and health, hold equal importance at 21%. The environmental dimension stands at 17%."
Author Response
Comments and suggestions from the fourth reviewer:
- “The manuscript delves into the potential of Medicinal and Aromatic Plants (AMP) within the realm of sustainable tourism, with a specific focus on mountainous regions. The author commendably underscores the socio-economic and environmental ramifications of AMPs, particularly against the backdrop of rural-to-urban migration and the recent Covid-19 pandemic. Nonetheless, the paper leans more towards a practical report than an academic article due to the absence of a clear development of research questions or hypotheses grounded in prior research.”
It's great to know that our efforts and dedication have been recognised by you. Your positive feedback is a great encouragement.
This paper aims to analyse the link between the aromatic and medicinal plants sector and the tourism sector. It does not have a research question or hypotheses, but it does have a literature background, research objectives, methods, results, discussion, implications and conclusions. The number of responses collected were not enough to do statistical inference, thus we decided to maintain this format of paper.
- “Introduction and Background:
- The manuscript offers a robust foundation on the importance of AMPs in sustainable tourism. Yet, a more exhaustive literature review would amplify the paper's depth.
- Clarity and Flow: Some segments are protracted and could benefit from brevity and restructuring for enhanced clarity.
Suggested Revision: "AMPs play a pivotal role in bolstering both local and national economies, simultaneously sustaining numerous rural livelihoods. They are integral to several industries, encompassing cosmetics, pharmaceuticals, and food."
"The AMP sector encompasses a spectrum of stages, from harvest to market. These stages encapsulate harvesting, processing, trading, value addition, and product innovation."
- Consolidate Information: There's a noticeable repetition and dispersion of certain points. Grouping related information would enhance coherence.
Suggested Revision: "Tourism, over the past half-century, has significantly augmented global wealth. Integrating AMPs can further accentuate this growth, ensuring quality offerings and catalysing business opportunities, especially in marginalized regions."”
We sincerely thank you for your observations and valuable comments on these sections. All your suggestions have been included in the new draft article (see it in a different colour).
- “Methodology:
- The methodology segment appears elusive or inadequately detailed. A comprehensive outline detailing research methods, data collection techniques, and analytical procedures is imperative.
- Detail the Data Collection Process: A meticulous breakdown of the data collection process is essential.
Suggested Revision:
- "The research team formulated a questionnaire, emphasizing open-ended queries to procure in-depth insights."
- "Google Forms was the chosen medium for data collection, complemented by an interview guide to maintain uniformity."
- "The survey witnessed participation from 25 producers/farmers."
- Elaborate on the Questionnaire**: The questionnaire, especially its open-ended components, requires further elaboration. The genesis of the survey questions and their sources should be elucidated.”
The research methods were content analysis of the farmers' responses. The questionnaire was developed in google forms, and the analytical procedures were drawn up in excel. Data was collected from April to June. They were then analysed in excel using content analysis.
No previous study has carried out similar research. This questionnaire was designed to obtain farmers' perceptions and feedback on the link between the aromatic and medicinal plant sector. The references that helped develop these questions were:
- Taghouti, I.; Cristobal, R.; Brenko, A.; Stara, K.; Markos, N.; Chapelet, B.; Hamrouni, L.; Buršić, D.; Bonet, J.A. The Market Evolution of Medicinal and Aromatic Plants: A Global Supply Chain Analysis and an Application of the Delphi Method in the Mediterranean Area. Forests 2022, 13, 1–13, doi:10.3390/f13050808;
- Kala, C.P. Medicinal and Aromatic Plants: Boon for Enterprise Development. J Appl Res Med Aromat Plants 2015, 2, 134–139, doi:10.1016/j.jarmap.2015.05.002;
- Kala, C.P. Commercial Exploitation and Conservation Status of High Value Medicinal Plants across the Borderline of India and Nepal in Pithoragarh. Indian Forester 2003, 129, 80–84, doi:10.36808/if/2003/v129i1/2238;
- Nicola, S.; Hoeberecht, J.; Fontana, E.; Saglietti, D. Medicinal and Aromatic Plants in Italy: Situation and Perspective for the Piedmont Region. Acta Hortic 2004, 629, 375–382, doi:10.17660/ActaHortic.2004.629.48.
- “Results and Discussion:
- The manuscript sheds light on AMPs' potential in revitalizing rural and economically challenged areas. The reference to AMPs during the Covid-19 pandemic is particularly captivating. Nonetheless, the results section would greatly benefit from a more organized data representation, preferably through graphs or tables.
- Question 3 (Q3), which reads, ‘What are the main barriers/obstacles that could hinder the development of the tourism sector linked to the AMPs?’ should be reframed to ‘What are the main benefits that could foster the development of the tourism sector linked to the AMPs?’
- Address Stakeholder Perspectives: The discussion should encompass the implications of the findings for diverse stakeholders, including farmers, policymakers, and investors.
The paper, given its topical relevance and insights, certainly has potential. However, it necessitates revisions in line with the aforementioned comments prior to being deemed fit for publication.”
Thank you for dedicating your time and expertise to reviewing our work. Your observations propel this work to a higher level of quality.
The results section presents 4 tables and 2 graphs.
The discussion and implications section aims to clarify and embrace several policy implications for stakeholders, farmers, investors and policymakers.
- “The manuscript is generally well-composed, but a meticulous proofreading is advised to bolster clarity and coherence.
Page 11:
- "Another measure that could be implemented is the creation of community gardens in rural areas based on the cultivation of AMPs to enhance cultural understanding, education, and learning about this type of cultivation."
- Reviewer's Note: The term "to foster a more efficient culture" was ambiguous. The revised sentence provides clearer intent.
- "It would be beneficial to establish a comprehensive management plan. Additionally, creating training and education centers that offer techniques for producers and entrepreneurs in the AMP sector can help mitigate negative impacts, such as overexploitation, deforestation, and other environmental challenges [10]."
- Reviewer's Note: The original sentence structure was intricate and potentially perplexing. The revised version breaks it down for enhanced clarity.
Page 10:
- "The results indicate that farmers prioritize the social dimension and brand image in both the first and third questions."
- Reviewer's Note: The repetitive use of "dimension" made the original sentence somewhat tedious. The revised sentence is more concise.
- "Other dimensions are also significant from the farmers' perspective. However, in the first question, both the economic and legal dimensions, as well as resilience and health, hold equal importance at 21%. The environmental dimension stands at 17%."”
We sincerely thank you for the time and expertise you devoted to evaluating our work. Your comments and suggestions were extremely valuable in improving the article.
Reviewer 5 Report
Comments and Suggestions for Authors
Tittle: Tourism of Aromatic and Medicinal Plants (AMP) and Sustainability: some practical evidences
Authors: João Capucho, Arminda do Paço and Pedro Dinis Gaspar
Line 57-58: That phrase it is not clear: It was also found that the mastery of AMPs was helpful when the coronavirus (Covid-19) pandemic appeared because of the lack of drugs and vaccines against this virus in the initial phase. Who found it? When the whole word had a problems with the virus nobody – on the beginning did not have anything.
Lines 64-65: promote the tourism sector; it was also found that 81 different species of AMP were simultaneously employed as a tourism product and in the treatment of malaria and typhoid. I think it is not for the whole tourism sector, it is rather for health tourism.
Lines 102-104: . Thus, in the AMP- based products sector, attention should be paid to several stages such as: (i) harvesting; (ii) processing; (iii) trading; (iv) value creation; (v) production; (vi) development of medicines, perfumes, cosmetic products, among others [4]. – the last ones they are not stages. Thay could have be used in different products – and that I think is correct but not in this sentence.
The lines: 134-137: defend that some AMPs are efficiently used to prevent the risk of contamination 134 and treat some symptoms of Covid-19, such as: (i) Quinquina; (ii) Eucalyptus; (iii) Thyme; (iv) Artemisia; (v) among others. These plants assume a very important role since they include bioactive constituents that can be applied in developing new drugs for the epidemic caused by the SARS-CoV-2 virus. It should also be noted that these drugs based on AMPs are examples of drugs with minimal or no adverse effects. For sure it is the true but how it is common to tourism sector? It is more suits to medical treatment.
3.1. AMPs and the Hospitality Industry – why it for hospitality? It should be rather for gastronomy sector including hospitality where gastronomy is.
Lines: 269-277 For example, tourist destinations can create gastronomic activities (show cooking; restaurant itinerary), fairs, and festivals. The main objective is to promote a product(s) related to AMPs and tourist destinations. In this sense, the main opportunity to promote the AMPs sector is the development of national and international fairs and events with the help of territorial marketing strategies. Thus, this type of marketing can produce several positive effects in attracting new tourists and encouraging the consumption and use of AMPs and products related to this raw material [7]. In sum, it remains the opportunity, as the one related to the growing trend of health and wellness tourism [10], for the creation and development of business in the AMP sector that the entrepreneurs should take – it is a huge mass, it like: ok, what more we can add… and put everything. Tourism – of course – is a huge basket with everything, but still you have to pick the right elements/products that it will be the right development of the region, etc.
The research which was made are very good but they are not fully related to the tourism – they very general. Maybe I would be better to leave some of the questions to some of remove. Because in the end is also the subject of sustainability which was not mention before. I am totally agree that the topic is right but in this case is a huge mess.

Author Response
Comments and suggestions from the fifth reviewer:
- “Line 57-58: That phrase it is not clear: It was also found that the mastery of AMPs was helpful when the coronavirus (Covid-19) pandemic appeared because of the lack of drugs and vaccines against this virus in the initial phase. Who found it? When the whole word had a problems with the virus nobody – on the beginning did not have anything.”
We would like to express our gratitude for the careful review and the valuable insights provided. Your contributions were instrumental in raising the quality of our article.
In response to the reviewer's suggestion, we can say that the aim of this paragraph is to express the usefulness and importance of aromatic and medicinal plants and medicines from them, during the lack of vaccination and some medicines at the beginning of the covid-19 crisis.
- “Lines 64-65: promote the tourism sector; it was also found that 81 different species of AMP were simultaneously employed as a tourism product and in the treatment of malaria and typhoid. I think it is not for the whole tourism sector, it is rather for health tourism.”
We appreciate the reviewer's comment. This paragraph has been edited because it does not deal with tourism in general, but about health tourism.
- “Lines 102-104: . Thus, in the AMP- based products sector, attention should be paid to several stages such as: (i) harvesting; (ii) processing; (iii) trading; (iv) value creation; (v) production; (vi) development of medicines, perfumes, cosmetic products, among others [4]. – the last ones they are not stages. Thay could have be used in different products – and that I think is correct but not in this sentence.”
Concerning the sentence mentioned in point 3, we have carefully reviewed it.
- “The lines: 134-137: defend that some AMPs are efficiently used to prevent the risk of contamination 134 and treat some symptoms of Covid-19, such as: (i) Quinquina; (ii) Eucalyptus; (iii) Thyme; (iv) Artemisia; (v) among others. These plants assume a very important role since they include bioactive constituents that can be applied in developing new drugs for the epidemic caused by the SARS-CoV-2 virus. It should also be noted that these drugs based on AMPs are examples of drugs with minimal or no adverse effects. For sure it is the true but how it is common to tourism sector? It is more suits to medical treatment.”
This section aims to describe this type of plant and its benefits. With this paragraph, the objective is to expose the benefits of these plants in preventing the risk of treatment and treat some symptoms. Obviously, countries that produce these plants could have an opportunity to create value and competitive advantages, especially through medical tourism.
- 1.“AMPs and the Hospitality Industry – why it for hospitality? It should be rather for gastronomy sector including hospitality where gastronomy is.”
We appreciate the reviewer's suggestion. We understand the concern raised about the sub-section: "AMPs and the Hospitality Industry" and why it should be hospitality and not gastronomy. We included the hospitality industry as a whole because of the many opportunities that hospitality companies have, such as: spa, aromatherapy, natural soaps, perfumes, cosmetics, many culinary plants, treating guests using AMPs (anxiety, insomnia, stress), among others. In fact, hospitality is wider, and can include as well the gastronomy.
- “Lines: 269-277 For example, tourist destinations can create gastronomic activities (show cooking; restaurant itinerary), fairs, and festivals. The main objective is to promote a product(s) related to AMPs and tourist destinations. In this sense, the main opportunity to promote the AMPs sector is the development of national and international fairs and events with the help of territorial marketing strategies. Thus, this type of marketing can produce several positive effects in attracting new tourists and encouraging the consumption and use of AMPs and products related to this raw material [7]. In sum, it remains the opportunity, as the one related to the growing trend of health and wellness tourism [10], for the creation and development of business in the AMP sector that the entrepreneurs should take – it is a huge mass, it like: ok, what more we can add… and put everything. Tourism – of course – is a huge basket with everything, but still you have to pick the right elements/products that it will be the right development of the region, etc.”
We're grateful for the reviewer's comment. We understand the importance of a clear and objective interpretation of what tourism can offer and who should be chosen. The paragraphs have therefore been revised.
- “The research which was made are very good but they are not fully related to the tourism – they very general. Maybe I would be better to leave some of the questions to some of remove. Because in the end is also the subject of sustainability which was not mention before. I am totally agree that the topic is right but in this case is a huge mess.”
We would like to sincerely thank you for your comments, suggestions and expertise in reviewing this manuscript. In fact it is difficult to separate some aspects, even so we will try to make some arrangements in order the clarify the study.
Reviewer 6 Report
Comments and Suggestions for Authors
Thank you for the opportunity to review your manuscript. It is an interesting and timely paper.
Here are some comments that may assist develop it further:
Content
Overall, some interesting points are made on a topic that is quite timely in the development of new forms of tourism, particularly around wellness. The focus on AMP provides a new perspective and makes some useful points which will add to the literature in this field.
If it was me, I'd re-work the manuscript, perhaps with some editorial assistance, to tighten the focus on AMP / tourism. I see some limitations in the data (see below) which limit the scope of the paper, and which could suggest to readers that you are making overly ambitious claims based on a small and specialized sample size, etc.
I'd also reconsider the material around sustainability. The evidence in the manuscript on 'sustainability' seems marginal to the main argument and the evidence presented.
A shorter, more focused manuscript presenting your findings on AMP / tourism could be a better idea. I'd also bolster the tourism references with further sources on wellness tourism and trends towards using AMP as part of the wellness offering at tourism destinations would add value and context to the paper, and weight to your conclusions.
Editorial suggestions
Line 2: I think the title could be made more specific and relevant by re-framing it as something like:
'Tourism and aromatic and medicinal plants: Lessons from Italy'
Line 9: Maybe add some specificity by adding 'in Italy' after '... industry'.
Line 17: Rather than 'merging', maybe re-word as something like 'more closely connecting'.
Line 19: Rather than ' discriminating', I suggest something like 'useful'.
Line 25: I suggest adding '$US' before '9.630 billion' if you are referring to $ American, or otherwise indicate what currency you are using.
Lines 29 - 30: Source for this statement? In fact this paragraph, which is the basis for arguing the importance of AMP, makes several statements that should be given sources.
Line 39: Same point
Line 42: Rather than use a numbered source, such as [3] here, I'd suggest you use the 'author's' name, then follow with the numbered ref. So here I suggest 'According to the World Bank [3], world trade in...'.
Line 61: Same point.
Line 108: Same point.
Line 134: Same point.
Line 142: Suggest deleting 'the' and 'developed by' in the last part of this line, so it would now read 'In various studies [4,14,20], the rural and ...'.
Line 156: Again, I suggest adding author's names ahead of the numbered source.
Line 167: Same point.
Line 170: Suggest amended heading 'The tourism sector and AMP'
Line 186: Suggest 'This' rather than 'the' at the start of the line.
Line 230: Again, I suggest adding author's names ahead of the numbered source.
Line 237: Same point.
Line 262: Same point.
Line 279 ff: I'd reiterate the location in this methods section. I'd also add the nature of the sample (farmers) and say something about the small sample size.
I note, too, the sample seems to be limited to farmers; did you consider sampling tourists or potential tourists or tourism operators?
Line 308: I suggest using 'responses' rather than 'answers' in Table 2
Also - I'd spell out the full sentence in the first category, instead of '(as ornamentals...)'
Line 309: Suggest 'Source: data collected as part of this study', or not have this at all as it will be obvious the data comes from your research.
Line 317 / 8: You have used the same title for this table as following Table (Table 4).
Again Tables 3 & 4 - suggest using 'responses', rather than 'answers'.
Line 334: Figure 1 title - maybe use a more descriptive title, like 'reasons to be more sustainable', rather the present 'frequencies of dimensions in question 1'.
Line 335: Same point as Line 309, above.
Line 337: Suggest a more descriptive title for figure 2, something like 'benefits of tourism linked to AMPS'.
Line 373: Add author's name ahead of the numbered source.
Line 380: Same point.
Line 429 ff: Conclusions. I'd add more material on the limitations in this study - specific location, timing, small sample size, specialized nature of the sample (farmers, not tourists, tourism industry...).
Line 459: Suggest 'would be created' be replaced with something like 'should be pursued' and replace 'and it would also facilitate the identification...' with something like 'which would also assist the development of an appropriate research agenda'.
Line 324 & 328 - same point as for Line 309, above.
Comments on the Quality of English Language
Good overall. The comments above may assist develop this further.
Author Response
Comments and suggestions from the sixth reviewer:
- Overall, some interesting points are made on a topic that is quite timely in the development of new forms of tourism, particularly around wellness. The focus on AMP provides a new perspective and makes some useful points which will add to the literature in this field.
We would like to express our appreciation for the review and the valuable comments and suggestions provided.
It's great to know that our efforts and dedication have been recognised by you. Your positive feedback is a great encouragement.
- If it was me, I'd re-work the manuscript, perhaps with some editorial assistance, to tighten the focus on AMP / tourism. I see some limitations in the data (see below) which limit the scope of the paper, and which could suggest to readers that you are making overly ambitious claims based on a small and specialized sample size, etc.
We appreciate your comment. Firstly, the questionnaire was developed within the framework of the PAM4Wellness project. Next, the link between tourism and aromatic and medicinal plants is poorly worked in the literature, there are automatically limitations in the construction and drafting of a questionnaire. Finally, the number of responses relates to the small and growing sector of aromatic and medicinal plant farmers in Portugal.
- I'd also reconsider the material around sustainability. The evidence in the manuscript on 'sustainability' seems marginal to the main argument and the evidence presented.
We sincerely appreciate your suggestion. The title has therefore been modified in order to be more related with all paper.
- A shorter, more focused manuscript presenting your findings on AMP / tourism could be a better idea. I'd also bolster the tourism references with further sources on wellness tourism and trends towards using AMP as part of the wellness offering at tourism destinations would add value and context to the paper, and weight to your conclusions.
We would like to thank the reviewer for his work and expertise in assessing this manuscript.
We would like to clarify an important point regarding the identification of the country covered in the study. The work in question focuses on Portugal, not Italy. I apologise if there has been any misunderstanding and thank you for the opportunity to clarify this detail.
Finally, thank you for your editorial suggestions, all of which have been analysed and modified (see the text in red colour).
Round 2
Reviewer 1 Report
Comments and Suggestions for Authors
1. In the four categories in Figures 1 and 2 of the article, farmers' answers should be classified into four parts instead of directly stating that the author divides them into four parts. How the four parts are connected with the farmers' answers is unclear.
2. In the discussion and implications, the author's discussion often has many connections with Kala, C.P. Can you add some more literature to discuss?
Comments on the Quality of English LanguageArticle writing can add a more rigorous academic writing style.
Author Response
Comments and suggestions from the first reviewer (Round 2):
- In the four categories in Figures 1 and 2 of the article, farmers' answers should be classified into four parts instead of directly stating that the author divides them into four parts. How the four parts are connected with the farmers' answers is unclear.
We would like to express our appreciation for the second review and the valuable comments and suggestions provided.
“Concerning the first and third questions inquired to farmers, the answers were selected and clustered into four dimensions: (i) Environmental; (ii) Economic and legal; (iii) Social and brand image; and (iv) Resilience and health. Figure 1 and Figure 2 show the frequencies of the dimensions in the farmers' answers to questions 1 and 3, respectively.” In other words, three questions were put to the farmers, and according to the answers obtained, both question 1 and question 3 were breached into four main dimensions. In this way, it was possible to find the frequency of each dimension and understand the perception of aromatic and medicinal plant farmers linked tourism sector of having more sustainable businesses and what the benefits are. In sum, a content analysis was followed to get the four dimension.
- In the discussion and implications, the author's discussion often has many connections with Kala, C.P. Can you add some more literature to discuss?
We appreciate your comment. However, one of the limitations of this study is the lack of studies and literature in the field of aromatic and medicinal plants - tourism, management, economics, among others. We would have liked to include more bibliography and strengthen this paper in terms of references, but everything that was found and analysed is presented in the article and is correctly explained, linked and referenced.
Reviewer 4 Report
Comments and Suggestions for Authors
The revisions made in response to my initial review have been executed commendably. I would like to further note that the development of the questionnaire items within the paper is thoughtfully grounded in references to farmers' perceptions and their feedback concerning the interconnections between the aromatic and medicinal plant sectors.
Wishing you the best of success with your dissertation.
Author Response
Comments and suggestions from the fourth reviewer (Round 2):
- “The revisions made in response to my initial review have been executed commendably. I would like to further note that the development of the questionnaire items within the paper is thoughtfully grounded in references to farmers' perceptions and their feedback concerning the interconnections between the aromatic and medicinal plant sectors.
Wishing you the best of success with your dissertation.”
We sincerely thank you for your positive and encouraging feedback. Your words are very motivating and appreciated.
Reviewer 5 Report
Comments and Suggestions for Authors
That version - and esspecially changing the titlle is much better then before. So in such shepe can be printed.
Author Response
Comments and suggestions from the fifth reviewer (Round 2):
- “That version - and especially changing the title is much better then before. So in such shape can be printed.”
We would like to express our gratitude for the careful review and the valuable insights provided. Your contributions were instrumental in raising the quality of our article.
Reviewer 6 Report
Comments and Suggestions for Authors
Thaks you for the opportunity to review your revised manuscript. I can see many improvements and clarifications which add to the value of the reported study.
I'm happy to recommend publication, though would some editorial changes to break the v long sentence at the start of the Methods section (Lines 277 - 282). Please forgive me for making such a bold suggestion, but perhaps this paragraph could be re-framed as something like:
'Data was collected as part of the PAM4WELLNESS project undertaken in Portugal. This study aimed to contribute to understanding the economic value of the endemic collected species already contributing to the cosmetic and pharmaceutical industry there. At present, the harvesting of these species is primarily aimed at forest clearing and fire prevention, not at extracting the potential of the collected material. In fact, at present the only value extraction from the harvest is through its dispatch to the biomass industry.
This project also aims to explore the sustainability of the harvesting and use processes. Through a suitable benchmarking exercise, the study aims to identify best practice for production processes, waste use and treatment, and leveraging other economic benefits such as tourism. All this is aimed at maximizing economic return while minimizing environmental impacts and reducing the ecological footprint of the processes.'
Thank you again for the opportunity to review and comment on your manuscript.
Comments on the Quality of English Language
Generally fine - see above for some suggestions for a key paragraph in the manuscript.
Author Response
Comments and suggestions from the sixth reviewer (Round 2):
- “Thanks you for the opportunity to review your revised manuscript. I can see many improvements and clarifications which add to the value of the reported study.
I'm happy to recommend publication, though would some editorial changes to break the v long sentence at the start of the Methods section (Lines 277 - 282). Please forgive me for making such a bold suggestion, but perhaps this paragraph could be re-framed as something like:
'Data was collected as part of the PAM4WELLNESS project undertaken in Portugal. This study aimed to contribute to understanding the economic value of the endemic collected species already contributing to the cosmetic and pharmaceutical industry there. At present, the harvesting of these species is primarily aimed at forest clearing and fire prevention, not at extracting the potential of the collected material. In fact, at present the only value extraction from the harvest is through its dispatch to the biomass industry.
This project also aims to explore the sustainability of the harvesting and use processes. Through a suitable benchmarking exercise, the study aims to identify best practice for production processes, waste use and treatment, and leveraging other economic benefits such as tourism. All this is aimed at maximizing economic return while minimizing environmental impacts and reducing the ecological footprint of the processes.'
Thank you again for the opportunity to review and comment on your manuscript.”
We would like to express our gratitude for the careful review and the valuable insights provided. Your contributions were instrumental in raising the quality of our article.
All the suggestions made were accepted and modified in the article.
Thank you very much for the time and effort you put into reviewing our article and thank you for your interest in accepting our article for publication.